# Adoption of Digital Vaccination Services: It Is the Click Flow, Not the Value—An Empirical Analysis of the Vaccination Management of the COVID-19 Pandemic in Germany

**DOI:** 10.3390/vaccines11040750

**Published:** 2023-03-28

**Authors:** Alexander Alscher, Benedikt Schnellbächer, Christian Wissing

**Affiliations:** 1BSP Business and Law School, Siemens Villa/Calandrellistraße 1-9, 12247 Berlin, Germany; christian.wissing@businessschool-berlin.de; 2College of Business and Economics, University of Hawaii at Hilo, 200 W. Kawili St., Hilo, HI 96720, USA; 3Faculty of Human and Business Sciences, Department of Economics, Saarland University, 66041 Saarbrücken, Germany; benedikt.schnellbaecher@uni-saarland.de

**Keywords:** COVID-19, pandemic, vaccination, vaccination services, platform technology, technology adoption model, acceptance model, innovation barriers

## Abstract

This research paper examines the adoption of digital services for the vaccination during the COVID-19 pandemic in Germany. Based on a survey in Germany’s federal state with the highest vaccination rate, which used digital vaccination services, its platform configuration and adoption barriers are analyzed to understand existing and future levers for optimizing vaccination success. Though technological adoption and resistance models have been originally developed for consumer-goods markets, this study gives empirical evidence especially for the applicability of an adjusted model explaining platform adoption for vaccination services and for digital health services in general. In this model, the configuration areas of personalization, communication, and data management have a remarkable effect to lower adoption barriers, but only functional and psychological factors affect the adoption intention. Above all, the usability barrier stands out with the strongest effect, while the often-cited value barrier is not significant at all. Personalization is found to be the most important factor for managing the usability barrier and thus for addressing the needs, preferences, situation, and, ultimately, the adoption of the citizens as users. Implications are given for policy makers and managers in such a pandemic crisis to focus on the click flow and server-to-human interaction rather than emphasizing value messages or touching traditional factors.

## 1. Introduction

### 1.1. Lack of Research in Vaccination Management

The COVID-19 pandemic, caused by the severe acute respiratory syndrome coronavirus type 2 (SARS-CoV-2), led to serious social and economic global disruptions, culminating in the largest global recession since the Great Depression (IMF, International Monetary Fund Data, 2022). Research has focused, so far, either on the optimization of traditional approaches (e.g., sterilization protocols, hygiene rules, testing regime, contact tracing, and quarantine) [1,2] or on the development of innovative approaches such as mRNA vaccines [3]. As the different infection waves have shown, innovative vaccines can only provide positive results, when people accept and use the related services given that vaccination processes offer the most effective protection against the pandemic [4]. Since not only the supply of vaccines is limited but also medical facilities, and healthcare workers are constrained as resource capacity, the vaccination management is also a general economic problem which requires an optimal allocation strategy by policy makers and pandemic officers [5]. Against this background, this study focuses on the digital management of vaccination processes, including services such as the registration and admission in the vaccination system, the booking of vaccination dates, postinjection monitoring, documentation and reminders to follow ups [6].

Xu et al. [7] differentiates six research clusters based on the analysis of 5070 research contributions on the COVID-19 vaccination: (1) attitudes towards vaccination (e.g., vaccination willingness and misinformation), (2) immunoinformatic analysis (e.g., vaccine design and DNA vaccines), (3) clinical research on vaccines (e.g., clinical trials, bioinformatics, and drugs), (4) vaccine effectiveness (e.g., immune response, immunization, and antibody response), (5) side effects (e.g., safety and surveillance), and (6) public management (e.g., public health and health policy). The management of vaccination processes falls into cluster six, whereby the use and potential of digital platform technologies for the vaccination management and their adoption by users have been rarely scrutinized [8,9]. However, successful vaccination campaigns do not depend only on the quality and effectiveness of the vaccines but also on their administration and handling processes, with state-of-the-art digital platform services that people know from their social lives [6,10,11]. To react rapidly and efficiently in a pandemic situation, digital tools are indispensable in the vaccination management, given the fact that more than 50% of the 3.4 billion smartphone users should have health apps installed on their phones [12]. In Germany, where this research study is located, almost 60% of patients use digital tools for searching symptoms and disease-related information (Bertelsmann Foundation, 2022). Adoption of digital tools is a key factor in public management in general and healthcare specifically, so that this research study defines this phenomenon as “digital vaccination management”. In a database search of Ex Libris Primo with the search strings “COVID-19 management” and “digital tools”, only 35 articles were found (see Section A.1), of which only three articles put a focus on diagnosis and treatment (which is also our research focus with vaccination management). These three studies present a detailed review of the role of artificial intelligence as a decisive tool for the prognosis, analysis, and tracking of the COVID-19 cases [13]; a proposal for public participation, digital solutions, and e-health initiatives [14]; and a description of an early warning, in-hospital, mobile risk-analysis app for diagnosing COVID-19 [15]. No articles on e-health tools for vaccination management were found.

The adoption rate is as a crucial success factor since the public vaccination campaigns against COVID-19 have shown how important it is to quickly build the necessary capacity for process and service innovation and to manage it in a functional way [16]. Another study from Chowdhury et al. [17], which discovered 415 COVID-19-related mobile health applications in 2021, showed that of the relevant 75 applications only 3–5% were designed to meet the health literacy and adoption needs of communities. Countless new technologies promote health service innovations, but they also force possible resistance on the user side [18]. This research study aims to understand the barriers and factors of adopting digital vaccination services by following the research question:

RQ: Which platform configuration works best to overcome adoption barriers and increase the adoption intention of digital vaccination services?

The paper is structured as follows. Based on the innovation acceptance and resistance theory, eight barriers are derived, which are then referred to the configuration categories of the TOPCOP patient portal model and formulated into a set of hypotheses in the theory section (Section 1). A web-based survey in Germany—approved and coordinated with the federal health ministry of Saarland—was conducted on the offered digital vaccination services. Based on 404 valid surveys, a component-based structural equation model is applied to analyze platform configuration domains (Section 2, Research Design). In the results section (Section 3), the effects of psychological, functional, and individual barriers on the intended adoption rate of digital vaccination services and the effects of the configuration categories on the barriers are shown, highlighting the significance of usability, security risk, and image barriers. The discussion section (Section 4) proposes an adjusted model for digital vaccination services, and implications for the policy makers managing the pandemic and health services are presented. Finally, the conclusions section (Section 5) emphasizes the key learnings for policy makers to focus on the usability barrier and personalization as the key factors for adoption of digital vaccination services.

### 1.2. Theory

#### 1.2.1. Barriers to Adopt Digital Platform Services

The acceptance of vaccination services can be explained by two research paradigms: innovation adoption (technology such as the acceptance model (TAM) [19,20], the theory of planned behavior (TPB) [21], the unified theory of acceptance and use of technology (UTAUT) [22], the dynamic acceptance model for the re-evaluation of technologies (DART) [23], and the unified theory of acceptance and use of technology 2 (UTAUT2) [24]), and the opposite, innovation resistance [25,26]. Though Ram [26] sees innovation resistance not as the opposite to innovation adoption, we perceive both research streams as valuable and relevant for serving our research question. Thereby, we follow the conclusion that “adoption of an innovation is conditioned by the overcoming of consumers’ initial resistance” [27] (p. 783). To answer our research question, we followed the more fine-grained resistance theory models and formed an adjusted digital vaccination adoption model (see Figure 1), which is mainly based on the work of Mani and Chouk [27].

Regarding functional barriers, the perceived value of a digital service plays a relevant role in determining the intention to adopt. Value barriers are often related to price–benefit ratios when adopting an innovation, usually referring to substitute products or services [28]. This also applies to the performance differences between the innovation and the substitute service [29], which is essential in healthcare systems without point-of-service payments. In this case, the notion of price should be understood in an extended way, according to transaction cost theory [30], as the sum of the efforts to register at the vaccination platform, and, according to the opportunity cost concept, as foregoing the benefits of other forms of vaccination (e.g., vaccination via mobile vaccination teams and vaccination in clinics). Another major resistance barrier is the usability of the innovation [31]: when the functionality of an innovation is difficult to understand or does not unfold, people tend to reject technological innovations [26]. If the users do not understand how to use the digital vaccination service, they bounce or break up the session. Close to usability, a valid concern of potential service users is the security risk of the personal information involved in the service interaction [32]. This security risk barrier is of major concern when adopting digital services [33], especially for vaccination services. Medical data are perceived as one of the most sensitive; thus, they are data of users most worth protecting.

In the context of psychological barriers, the image barrier describes how people associate innovations with known identities [31]. Such a self-image incongruence can impede the innovation acceptance when the identity does not fit to the proposed function of the innovation [34]. Such a mismatch takes place if individuals connect the vaccination service, or healthcare in general, with personal interactions and consequently shun digital solutions. Moreover, daily routines and habits can conflict with digital service innovations such as the automated, unpersonal vaccination registration. These tradition barriers can have a negative influence on the acceptance of an innovation or can even lead to a complete rejection of an innovation [34]. In the last decades, technological developments have radically changed the behavior, attitudes, and beliefs of individuals. The phenomenon “technological vulnerability” became a challenge for research in digital domains [35] and has been analyzed in terms of the two barriers: “technology anxiety” and “perceived technological dependence” [27]. A general technology anxiety might lead to an aversion to new digital approaches for managing healthcare activities, such as digital vaccination services, due to a common fear of individuals to use them [36]. The perceived dependence barrier emerges because of the phenomenon that more and more areas of life are unthinkable without the reliance on technology. This can easily be seen as dependence, leading to dire results when the technology might be, due to unforeseen circumstances, not available anymore [37]. Due to the safety sensitivity of the healthcare data in the vaccination service, any digital substitution might raise doubts about this potential dependence and, in turn, negatively affect the intention to adopt a new digital service.

Individual barriers relate to the status quo bias (SQB) theory, which claims that maintaining the status quo (i.e., inertia) is an individual variable leading to resistance [38]. This personal predisposition leads an individual to prefer the current situation to situations of uncertainty or change [25]. Hence, inertia increases resistance to the new digital vaccination service.

In summary, we conclude the following hypotheses on the effect of adoption barriers:

**H1.1.** 
*The value barrier (perceived price) negatively affects the intention to adopt the digital platform service.*


**H1.2.** 
*The usability barrier (perceived complexity) negatively affects the intention to adopt the digital platform service.*


**H1.3.** 
*The perceived security risk barrier negatively affects the intention to adopt the digital platform service.*


**H1.4.** 
*The image barrier (self-image incongruence) negatively affects the intention to adopt the digital platform service.*


**H1.5.** 
*The tradition barrier (need for human interaction) negatively affects the intention to adopt the digital platform service.*


**H1.6.** 
*The perceived dependence barrier negatively affects the intention to adopt the digital platform service.*


**H1.7.** 
*The technology anxiety barrier negatively affects the intention to adopt the digital platform service.*


**H1.8.** 
*The individual inertia barrier (status quo bias) negatively affects the intention to adopt the digital platform service.*


#### 1.2.2. Platform Configuration of Vaccination Services to Mitigate Adoption Barriers

Though healthcare providers have been sluggish in adopting new technologies, the COVID-19 pandemic has accelerated the use of digital technologies in healthcare [39]. For instance, the German government reinforced the use of digital services in fighting the COVID-19 pandemic by developing the “Corona Warn App” (for contact tracing in 2020) or by issuing digital vaccination certificates (since 2021). The literature called for quick and, above all, unbureaucratic (digital) approaches with appropriate incentives along the vaccination process [2]. To establish and design successful digital services such as those on health platforms, their architecture and management are decisive points [40]. The TOPCOP taxonomy on patient portals serves health information managers to classify and structure digital health platforms [41]. For offering digital vaccination services, the original TOPCOP patient portal was reduced in its complexity to three platform configuration areas: personalization, communication, and data management (see Section A.2).

#### 1.2.3. Personalization to Facilitate User Perception and Lower Adoption Barriers

Personalization plays a crucial role in optimizing interactions between service providers and users; thus, it is a growing management trend [42]. Personalization, understood as the ability of an organization to customize its services according to one’s own needs and preferences [43], increases the viability of the digital vaccination service for the user and thus should lower related notions of innovation. Moreover, it increases the perceived value of the new vaccination service and eventually lowers the value barrier so that a potential user finds the proposed innovation of the vaccination service attractive [27]. Users with the option of personalization will use it to increase one’s own usability and to overcome the fear of security risks. Pearson et al. [44] cite personalization as one of the drivers of online-based usability. Finally, the barrier of technology anxiety relates to an irrational fear when confronting an innovation such as the digital vaccination service [36]. Most anxiety management approaches rely on letting individuals engage with the object of the anxiety in a limited, controlled fashion. For the digital vaccination service designed for usability, we expect a mitigating impact of personalization on technology anxiety. To sum up, we propose:

**H2.1.** 
*Personalization reduces the usability barrier (perceived complexity) in the digital platform service.*


**H2.2.** 
*Personalization reduces the value barrier (perceived price) in the digital platform service.*


**H2.3.** 
*Personalization reduces the security risk barrier in the digital platform service.*


**H2.4.** 
*Personalization reduces the technological anxiety barrier in the digital platform service.*


#### 1.2.4. Communication Providing Targeted Information to Mitigate Adoption Barriers

An adequate information basis is required to mitigate any suspicions concerning new technologies or service innovations [45]. In terms of the offering of the digital vaccination service, its potential and outcome have to be evaluated by individuals and compared to the status quo, which requires a high level of transparency of the potential changes [46], in particular in the risk-avoidance culture of the healthcare context [47]. Heidenreich and Kraemer [48] demonstrated that employing mental simulation can reduce innovation resistance and, accordingly, information might alleviate biased perceptions regarding the image of the digital vaccination service and dispel prior stereotypes. In general, innovations are associated with certain identities, such as a product category, brand, or country of origin [29], which might foster innovation resistance [31]. The image barrier is triggered when identities run counterpoint to the substituted product or context. In the case of health services, the stereotyped insecure digital services and traditional healthcare interactions, often romanticized as trust-based [49], could such counterpoints. Closely connected to this, the tradition barrier can occur, referring to the fear of changes in socially learned values, beliefs, behaviors, and routines [50]. If a tradition, such as the need for human interactions in healthcare systems, is questioned by a new, anonymous digital service, communication might trigger sociopsychological processes to overcome this barrier and thus promote the adoption of the innovative vaccination service [51]. On a deeper level, the provision of information concerning the involved service processes might reduce technological vulnerability, perceived technological dependence as well as technological anxiety [27]. According to the attribution theory, the perceived technological dependence is an attributional mechanism that enables people to experience and express their skepticism about a new technology-based service, such as for the vaccination. Ratchford and Barnhart [52] (p. 1212) describe this perception as “a sense of being overly dependent on, and a feeling of being enslaved by, technology”. Communicating targeted information about the object of the technological anxiety, i.e., the vaccination service, is described as one possible path to mitigate these fears by giving the individual the possibility to engage with it on a cognitive level. While digital services today differ from physical-centered computer hardware, a study of 187 participants by Igbaria and Chakrabarti [53] found that education was able to significantly reduce computer-related anxiety and inertia of students, indicating a positive impact of structured information. To sum up, we propose the following hypotheses:

**H3.1.** 
*Communication mitigates the image barrier (self-image incongruence) in the digital platform service.*


**H3.2.** 
*Communication mitigates the tradition barrier (need for human interaction) in the digital platform service.*


**H3.3.** 
*Communication mitigates the perceived technological dependence barrier in the digital platform service.*


**H3.4.** 
*Communication mitigates the technological anxiety in the digital platform service.*


**H3.5.** 
*Communication mitigates the inertia (SBQ) barrier in the digital platform service.*


#### 1.2.5. Data Management to Ensure Service Speed and Reduce Adoption Barriers

Users of innovations are easily deterred by reactions that do not conform to their expectations regarding typical interaction patterns or quality levels of the chosen innovation [54]. Insufficient capacity is detrimental to the success of service providers [55], where capacity of a provider is defined as “highest quantity of output possible in a given time period with a predefined level of staffing, facilities, and equipment” [55] (p. 26). Different from physical services, instant responses are expected for digital services because the offer is compared to heavily standardized and AI-supported services by industry leaders, such as search engines or social networks. Perceived delays or mistakes are not as easily forgiven, especially if users disclose and submit sensitive healthcare information. A systematic data management is required to ensure service speed and short reaction times and thus avoid the fostering of adoption barriers against the vaccination service. This is supported by an analysis of the responses of 453 Korean healthcare service users, finding that an interactive quality was deemed a significant factor for patient satisfaction [56]. A reliable and speedy response is a precursor for the perceived value of digital services [57]. Sound data management should ensure a higher sense of usability and a lower sense of experienced security risk due to quick fixes, when the need might arise, and general responsiveness. Reliable data management should alleviate technological anxiety since smooth interaction reduces the risk of unforeseen events due to errors and worrying thoughts during waiting times. In summary, we propose the following hypotheses:

**H4.1.** 
*Data Management reduces the value barrier (perceived price) in the digital platform service.*


**H4.2.** 
*Data Management reduces the usability barrier (perceived complexity) in the digital platform service.*


**H4.3.** 
*Data Management reduces the security risk barrier in the digital platform service.*


**H4.4.** 
*Data Management reduces the technological anxiety in the digital platform service.*


## 2. Research Design

### 2.1. Federal State with Highest Vaccination Rate in Germany as Research Area

To answer our research question of which platform configuration works best to overcome adoption barriers, the federal state with the highest vaccination rate in Germany, i.e., Saarland, was chosen (see Table A1 and Figure A1). Each federal state in Germany had to build up vaccination services according to the “National Vaccination Strategy” from the German government in October 2020. The federal state of Saarland decided on a digital vaccination platform for the approximately one million inhabitants and chose the company samedi after a selection process on 27 November 2020. The specialized company, samedi, founded in 2008 in Germany, is a leading provider of e-health software for patient portal and care management solutions which is used by more than 40,000 individual healthcare providers for the coordination over 30 million patients in Germany, Austria, and Switzerland (www.samedi.com (accessed on 23 March 2022)). The digital health platform, samedi, was customized for the vaccination process with the top-level domain “www.impfen-saarland.de (accessed on 23 March 2022)”, further load-balancers to manage the traffic, and configurations to manage the following services: (1) vaccination appointment booking (registration), (2) invitation e-mail (with QR code) and short message reminder, (3) security QR code check-in, (4) admission management (check-in forms), (5) waiting list with data security compliant patient calls, (6) vaccination cabin allocation, (7) monitoring list, (8) vaccination documentation with vaccination batch and employee code scan, and (9) government data reporting. Nevertheless, the available vaccination slots were instantly booked, which led to stress, frustration, and anger of the citizens. For this reason, a vaccination preregistration list was set up and implemented in January 2021. Citizens registered with their preferences (i.e., vaccination location, daytime, weekday, and partner code), and after collecting registrations for two weeks, all registrations were randomized and, according to a smart algorithm, allocated according to the given preferences. The first randomization allocation took place on 27 January 2021, with an official notarization so that a fair and just allocation process was guaranteed. In April 2021, the German government amended the vaccination strategy and included 65,000 medical doctors in the vaccination process who vaccinated independently in their own practices. From the start of vaccinations in December until April, Saarland ranked number one in Germany in terms of vaccination rate (see Section A.3).

### 2.2. Understanding the Adoption by Means of a Survey of Vaccinated People

For the evaluation of the digital vaccination management, the public administration of the federal state of Saarland was approached with the research project together with the University of Saarland on 8 June 2021. The survey was approved by the federal minister of health on 12 August 2021 and went until 30 September 2021. The survey was set up with the software Unipark (Version EFS Survey 2021) and was advertised in the vaccination centers of Saarland with posters and flyers. The survey was anonymous and voluntary and consisted of 46 questions (see Section A.4 for the original German version and Section A.5 for the translated English version). The survey started with nine questions about control and context factors (e.g., gender, age, and education) and then continued for the items measuring the evaluation of the platform services and intention to adopt future digital health services. Respondents indicated their approval to the statements on a seven-point Likert scale, from totally disagree to totally agree (i.e., scale scores of 1–3 are negative, 4 is neutral, 5–7 are positive). For the operationalization of the model, the dependent variable of “adoption intention” is defined as the intention to adopt digital health services and is operationalized using a reflective, first-order construct, adapted from Heidenreich et al. [58] for the healthcare context. Following the approach of Mani and Chouk (2018), it is categorized in functional, psychological, and individual barriers, which in turn are operationalized in reflective, first-order constructs for their respective barriers. To assess communication provision, the operationalization of Auh et al. [59], in the form of a reflective, first-order construct, was utilized to estimate how expedient the information provision of the service was perceived. Analogous, personalization is based on the scale of Burnham et al. [60], which gauges the degree of personalization in a service. Data management reflects the capability of the digital platform to possess enough resources to react reliability and swiftly to user demands [61]. The items of both constructs are reflective and are united in first-order constructs.

The basis population is exceptionally well suited to address the research question because the German population was instructed early on to use the technology to gain access to the COVID-19 vaccination. Thus, the sample is less subject to the pro-change bias prevalent in innovation studies, namely, that mainly individuals participate in change and innovation with a positive attitude [46]. Data were collected of 404 valid respondents (see Appendix A). Strict confidentiality of the answers was guaranteed to the participants, mitigating the risk of a social desirability bias. To reduce the probability of common method bias, a full collinearity assessment approach was chosen to check for pathological collinearity. The results of the variance inflation factors (VIFs) of the full collinearity test were below 3.3. Hence, the findings indicate a low probability of common method bias in the model. For assessing the research model and the associated hypotheses, the dataset was analyzed using a component-based structural equation model due to its advantages concerning the sample distribution and size [62]. The model was calculated using SmartPLS 3.0 with path-weighing calculations and a case-wise replacement missing algorithm. A nonparametric bootstrapping with 5000 replications and individual level changes were used for the standard error calculations [63].

## 3. Results

The survey was answered by 411 participants (with 404 valid responses) in an average time of 12 min, whereof two-thirds of the respondents was female and one-third was male. The age distribution is distributed with 40% under 46 years, 50% between 46 and 67 years, and 10% older than 68 years. A total of 26% of the respondents were healthcare workers and 50% had a university degree. The vaccination reason, based on the national vaccination categorization, was 40% health-related and 35% because of age. In terms of satisfaction with the federal state of Saarland as the vaccination provider, 86% were positive or neutral and 85% would also (positive/neutral) use digital health services in the future. A total of 10% found the digital vaccination platform not easy to use, but also 11% classified themselves as biased with technological anxiety. Regarding digital platform configuration areas, 83% (positive/neutral) considered the platform appropriate for their concern (personalization), 83% (positive/neutral) for keeping them well-informed (communication), and 76% (positive/neutral) for being quick responding (data management). Most of the respondents very highly appreciated the digital vaccination platform in the configuration areas of personalization (by 52%) and communication (by 53%) (see Table 1).

To validate our adoption model, the quality of the measurement constructs was evaluated by testing them without structural connections. An exploratory principal component analysis was then conducted for the constructs. The indicator loadings consistently exceeded 0.708, suggesting indicator reliability for all constructs. Moving on, Cronbach’s alpha was computed for the constructs to determine the respective construct’s reliability. The results surpassed 0.7, indicating adequate reliability [64]. Examining convergent validity and the discriminant, the average variance extracted (AVE) was calculated, which surpassed the minimum level of 0.5. The comparison with the squared intercorrelation of the constructs also resulted in adequate results [65]. Looking at the structural model, the path coefficients and significances according to the proposed research model were computed (see Table 2).

The measured R^2^ demonstrates an appropriate fit between the data and the model, with values ranging from 0.10 to 0.51. Contemplating the presence of multicollinearity at the structural level, the VIFs were estimated, but all fell in the required parameters of below five. Eventually, the model fit was assessed by calculating the standardized root-mean-square residual (SRMR = 0.08). While the quality of this indicator is not yet fully verified for PLS-based structural equation modeling, the value is adequate with the more conservative threshold for covariance-based structural equation modeling. For the first set of hypotheses, we tested the functional, psychological, and individual barriers on the intention to adopt (ItA) digital vaccination services (see Table 3).

From the eight adoption barriers, only the usability barrier (*t*-Stat 5.6, *p* < 0.00), the security risk barrier (*t*-Stat 2.6, *p* < 0.01), and the image barrier (*t*-Stat 5.5, *p* < 0.00) could be proven to be statistically significant.

The hypotheses about the platform configuration areas were tested as the construct factors for personalization, communication, and data management were measured on the adoption barriers (see Table 4).

Based on the given results, personalization only has a significant effect on the usability barrier (*t*-Stat 5.8, *p* < 0.00) and on the technological anxiety (*t*-Stat 2.2, *p* < 0.04), whereas the influence on value and perceived dependence is not strong enough, though the direction shows as expected by the model. Communication has a significant correlation with the image (*t*-Stat 2.4, *p* < 0.02), perceived dependence (*t*-Stat 3.1, *p* < 0.01), and technological anxiety (*t*-Stat 2.1, *p* < 0.04) barriers but no significance with the tradition and inertia barriers. Moreover, a significant effect of communication was found on the usability barrier (*t*-Stat 2.4, *p* < 0.02). The directions of the effects are all negative, as expected by the model. Data management has significant effects on the usability (*t*-Stat 4.7, *p* < 0.00) and technological anxiety (*t*-Stat 2.8, *p* < 0.01) barriers but no significance with perceived value and security risk barriers. A significant effect of data management was also found on the image barrier (*t*-Stat 2.4, *p* < 0.02). For an overview of the significant results, see Figure 2.

## 4. Discussion

### 4.1. Platform Services for Vaccination Processes—Implications for Further Research

The resistance model of Mani and Chouk [27] has been originally developed for measuring the acceptance of new technologies in consumer-goods markets, so this study proposed an adjusted adoption model explaining platform services for vaccination services,, and in general, for new digital health services. This study gives empirical evidence for the applicability of the model. Eight barriers were identified that coin the acceptance of digital vaccination services based on the three platform configuration areas: personalization, communication and data management (see Figure 2). Most of the factor loadings confirm the model. Above all, the usability barrier has the highest effect on adoption intention and should be marked as the top influence factor. This study eventually supports the results of Ram and Sheth [31]. The significant image barrier, as the second most important factor, substantiates the findings of, for example, Laukkanen [34], stating that if the identity does not fit to the proposed function of a digital innovation, individuals may shun it. However, image, especially in the context of marketing communication, is often understood as a network of different associations from which expectations arise, which are not necessarily linked to one’s own self-concept [66]. Further studies are needed to clarify the role of image on innovation acceptance. Interestingly, only image has a significant effect in our model, and neither tradition nor technological vulnerabilities (perceived dependence and technological anxiety).

The insignificant role of value is quite striking. This stands in contrast to studies covering consumer-goods markets, in which the perceived value, as a lack of monetary and performance value of an innovation, has been shown as one of the key drivers in terms of adoption [29]. In addition, studies on digital technologies tackling COVID-19 showed the expected performance (as a proxy for value) as the strongest factor regarding technology adoption [18]. A possible reason is that the benefit of the vaccination is utmost high (given the risk of death from COVID-19 and given the regained freedom of economic activity) and that there is no substitute available (though COVID-19 testing may be a temporary substitute). Hence, the value in the original sense of a price–benefit ratio plays a minor role here. An alternative reason is related to the characteristics of the mandatory German healthcare system, where prices for treatments, drugs, vaccines, etc., are not shown to the patients and thus are not anchored in patients’ minds.

Though inertia, has shown to be a leading factor influencing the adoption of innovations [38], there has been no significant effect found in this study. During the COVID-19 pandemic, there have been repeated calls from politicians and experts to motivate the latecomers and vaccine defaulters to vaccinate to significantly increase vaccination rates. This inertia in terms of vaccine hesitancy [67] is not reflected in our study. It remains to be tested in future studies whether this result can be replicated in the healthcare system in general or whether it can be attributed specifically to, for example, the subjective perceived lethality of COVID-19 in the general population [68].

In addition, our study shows remarkable results on the influence of age and education on the adoption of digital platform services. Especially as many older people had to be vaccinated first (categorization in Germany started with the older 80 years old citizens in contrast to other countries such as the United States) where human service, tradition, and technological anxiety would have been to be expected quite relevant, it was shown that even 80% of the vaccination booking were completed online as the vaccination started in Saarland [69]. Assumingly, older people were supported by family or care personal, but the results from our study have shown that 85% booked by themselves online. Supporting this finding, age as a control factor did not influence the adoption intention. Nevertheless, we consider it necessary to investigate in further research what level of digital literacy is needed among vulnerable groups, such as the elderly, people living in rural areas or in developing countries, to facilitate these groups’ access to digital health services in general and digital services for vaccination management. Moreover, and in line with the meta-analysis by Byrnes et al. [70], which showed that men are more likely to take risks than women, our study also had a negative relation form gender on adoption, confirming the women were more restrictive to further use of digital platform services. Interestingly, the control factor of education also had a negative effect on intention to adopt, showing that a higher education tends to lead to a more conservative and careful behavior regarding the future use of platform services. Further research needs to clarify the counter-intuitive finding since professions with high educational requirements play a key role in healthcare systems and women constitute the majority workforce in many healthcare systems.

### 4.2. Platform Services for Vaccination Processes—Implications for Management and Policy

For pandemic managers and policy makers, an important question is how to handle a pandemic and how to maximize the rapid adoption of related health services. This study proposes a digital vaccination service model as a helpful approach to manage vaccination processes. When introducing those services, the configuration areas of personalization, communication, and data management are useful concepts to be considered, since all had a significant effect on the adoption barriers. Though not all barriers were significant for the adoption intention of vaccination candidates, the compound effect of all barriers may not be neglected.

According to our study, functionality and ease of use are of major importance. The usability had the highest effect on the adoption intention and should be marked as the top influence factor for configurating platform services. The click flow (see Section A.6) seems decisive to determine the conversion or bouncing of users on digital health platforms. Therefore, following a patient-centered approach, the guidance throughout the digital vaccination journey is an eminently important design and management aspect. The influence of personalization on the usability is the strongest in the whole model. Thus, personalization should be priority number one for policy makers and pandemic-fighting officers—the ability to structure a service according to the needs and preferences of the subjects increases its viability for the user and thus should lower related notions of innovation resistance [71].

Communication, as the second area of platform configuration, has the highest number of effects on the adoption barriers, namely, usability, image, perceived dependence, and technological anxiety. As healthcare industries display a dominant risk-avoidance culture [47], information processing on platform services is critical to the vaccination management. This concerns information preparation as well as its distribution and updating. Detailed information enables individuals to judge the value of an innovation since its characteristics become understandable and the comparison to the detrimental status quo is possible [46]. Communication provides targeted information to lower barriers (i.e., usability), to map user preferences with the platform setup (i.e., addressing the image barrier), and to reduce the fear of technological vulnerabilities (i.e., minimizing technological dependence and anxiety barriers). It has a high influence on the psychological perception of how the digital service is accepted or rejected. In the federal state of Saarland, the information process was delivered not only on the federal health ministry site but also on the special site, www.impfen-saarland.de (accessed on 23 March 2022), with explanation and guidance, as well as an ongoing process with e-mail reminders, information attachments, route direction, SMS reminders, and a check list for the vaccination process in the dedicated vaccination centers.

Data management, as the third area of platform configuration, had an impact on usability and technology anxiety, which have already been shown in the literature about user churn due to insufficient capacity [55]. The management of vaccines and the management of related services must be thoughtfully differentiated as their perceived values have different sources of success. Partnerships between government organizations and the private sector, such as public–private partnerships or outsourcing models, can promote higher vaccination rates. For instance, the federal state of Saarland did not rely on their servers but hired a private e-health company with extensive experience and expertise. Moreover, with the engagement of this private partner, the federal state of Saarland followed an agile data management approach, including the setup of an additional upstream infrastructure with extra load balancers in front of the digital applications, such as scheduling and documentation, to ensure a smooth operation and a speedy and responsive experience with the digital vaccination platform. This approach is also in line with recent research calls to proactively promote organizational agility and flexibility to be able to act quickly in the event of unforeseen crises or changes [72].

## 5. Conclusions

### 5.1. Limitations

This study is located in Germany, an industrialized, developed country where enough vaccination doses can be ordered without any significant financial restrictions. Other countries and societies differ from this capability and are additionally coined by different social and cultural patterns. For instance, a striking difference was discovered between value and performance appreciation in the developing country of Nigeria [18] and the usability appreciation as top factor in Germany. Even in Europe, pandemic management was executed in different ways. Future research may even disclose differences in the countries. Another limitation is the focus on the COVID-19 vaccination campaign, which might be different for other (public) health services with less media attention and with less perceived threat and social restriction. In addition, the timing of this study is also special since it represents a comparatively large number of vaccination candidates who decided to be vaccinated comparatively early. At this point, we recommend the investigation of the groups of persons who decided for a later vaccination or who are vaccine-hesitant.

### 5.2. Summary

This study bridges existing adoption and acceptance theories and models from consumer markets to the public health sector and delivers conceptualization and guidance for policy makers and pandemic managers. Based on the presented adjusted adoption model, policy makers have a framework of the challenges and barriers in managing public health services, such as vaccination offerings. As a further contribution, this study provides an orientation on how to prioritize the right platform configuration for digital health services. In this sense, we highlight three key learnings: (1) The usability barrier is the most important barrier that harms adoption, whereas value does not play a dominant role in contrast to consumer markets. This barrier can be managed by all three digital platform configuration areas of personalization, communication, and data management. (2) Personalization is the most important factor for managing the usability barrier by optimizing the best click flow on the digital health platform to address the needs, preferences, and situations of the citizens as users. (3) Inertia is often stressed as a problem within public discussion. It is said that latecomers and vaccine defaulters harm positive vaccination rates. Future research may evaluate these findings and the overall adjusted adoption model for digital platform services to manage COVID-19 vaccination in other organization forms and cultural areas to contribute to the goal of improving the management of future pandemic settings.

## Figures and Tables

**Figure 1 vaccines-11-00750-f001:**
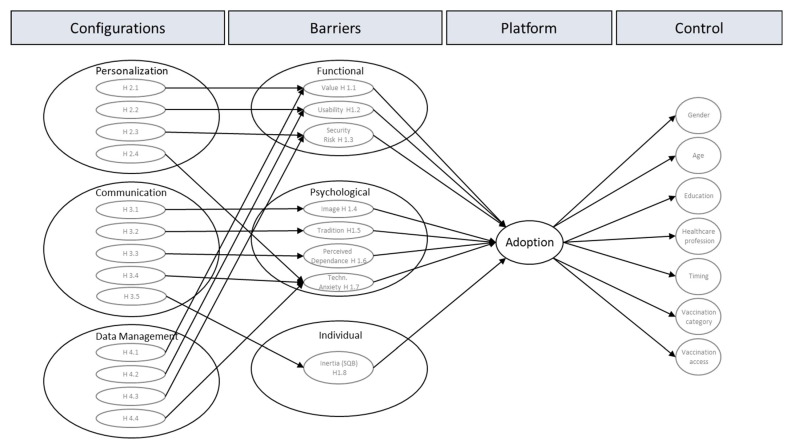
Adoption Model for digital platform services to manage COVID-19 vaccination (based on Mani and Chouk [27]).

**Figure 2 vaccines-11-00750-f002:**
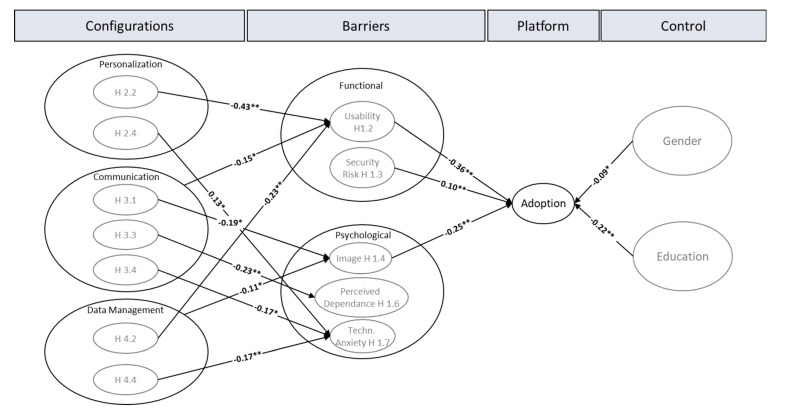
Significant factors in the adoption Model for digital platform services to manage the COVID-19 vaccination. Notes: ** *p* < 0.01, * *p* < 0.05.

**Table 1 vaccines-11-00750-t001:** Descriptive Results of Digital Vaccination Management Survey.

Variable	Description	M	SD	Explanation
Duration	Time to fill out survey.	706	435	Average was 11.8 min (3.6 min–55 max)
Age	(1–4) <46, (5–6) 46–67, (7–8) >67	4.69	1.64	40% 18–45, 50% 46–67, 10% >67 years
Gender	(1) male, …, (5) female	3.67	1.88	32% male, 66% female
Education	(1–5) school/job, (6–7) B./Master	4.84	1.53	49% school/job, 41% bachelor’s/master’s
Healthcare w.	1 (yes), 2 (no)	1.74	0.44	26% yes, 74% no
Priorisat. Group	(1–4) age, (5) care, (6) health, (7) job	4.91	1.59	35% age, 15% care, 40% health, 8% job
Satis_3	Satisfied with vaccination process	5.61	1.89	78% positive, 8% neutral, 14% negative
Intent_Adopt_2	intend to use dig service in future	5.45	1.88	72% positive, 13% neutral, 15% negative
Usabilit_Barr_1	digital portal was easy to use	1.95	1.62	86% positive, 4% neutral, 10% negative
Te_Anx_Barr_2	not using d. tech. to avoid errors	1.67	1.29	11% agree, 9% neutral, 80% disagree
Personal_1	portal appropriate for concern	5.65	1.86	77% positive, 6% neutral, 16% negative
Commun_1	portal keeps me well informed	5.63	1.99	76% positive, 7% neutral, 15% negative
Data_R_Man_1	portal responded quickly	4.88	2.04	64% positive, 12% neutral, 24% negative

Notes: Healthcare w. = healthcare worker; Priorisat. Group = prioritization group, according to National Vaccination Guideline; Satis_3 = satisfaction variable 3; Intent_Adopt_2 = adoption intention variable 2; Usabilt_Barr_1/2 = usability barrier variable 1/2; Te-Anx_Barr_2 = technology anxiety barrier variable 2; Person_1 = personalization (configuration) variable 1; Commun_1 = communication (configuration) variable 1; Data_R_Man_1 = data and resource management (configuration) variable 1_.

**Table 2 vaccines-11-00750-t002:** Results from Structural Equation Modeling.

	Image Barr	Individual Inertia	Perceived Dependence B.	Perceived Value Barrier	Security Risk Barr	Technology Anxiety Barr	Tradition Barr	Usability Barr	Intention to Adopt
Communication	−0.195	−0.110	−0.234	−0.088	0.006	−0.167	0.046	−0.153	
Data Management	−0.109	−0.006	−0.024	−0.444	−0.092	−0.172	0.113	−0.233	
Personalization	0.134	−0.199	0.128	−0.186	−0.127	0.131	0.097	−0.429	
Intention to adopt	−0.248	0.028	−0.104	−0.262	0.101	0.080	−0.046	−0.361	
Age									−0.02
Education									−0.22
Gender									−0.09
Vaccination Timing									0.01

**Table 3 vaccines-11-00750-t003:** Results from hypothesis H1 set about adoption barriers.

Hypothesis	Structural Relation	Original Sample	M	SD	*t*	*p*
H1.1	Usability Barrier (Rev) -> ItA	−0.361	−0.354	0.064	5.616	0.000 ***
H1.2	P. Value Barrier -> ItA	−0.262	−0.191	0.191	1.375	0.169
H1.3	Security Risk Barrier -> ItA	−0.101	−0.102	0.038	2.629	0.009 **
H1.4	Image Barrier (Rev) -> ItA	−0.248	−0.249	0.045	5.454	0.000 ***
H1.5	Tradition Barrier -> ItA	−0.045	−0.045	0.043	1.037	0.300
H1.6	P. Depend. Barrier -> ItA	−0.104	−0.103	0.065	1.596	0.111
H1.7	Tech. Anxiety Barrier -> ItA	0.080	0.080	0.054	1.469	0.142
H1.8	Individual Inertia -> ItA	0.028	0.009	0.043	0.657	0.511

*Notes*: ItA = intention to adopt; P. = personal; Tech. = technology; M = mean, SD = standard deviation, *t*/*p* = *t*-/*p*-value, * *p* < 0.05, ** *p* < 0.01, *** *p* < 0.00.

**Table 4 vaccines-11-00750-t004:** Results from the H2, H3, and H4 sets about effect on adoption barriers.

Hypothesis	Structural Relation	Original Sample	M	SD	*t*	*p*
Customization						
H2.1	Usability Barrier (Rev)	−0.427	−0.426	0.074	5.799	0.000 ***
H2.2	Perceived Value Barrier	−0.186	−0.148	0.135	1.376	0.169
H2.3	Security Risk Barrier	−0.127	−0.120	0.073	1.743	0.082
H2.4	Techn. Anxiety Barrier	−0.167	−0.128	0.060	2.175	0.030 *
**Communication**						
H3.1	Image Barrier (Rev)	−0.195	−0.191	0.082	2.385	0.017 *
H3.2	Tradition Barrier	0.046	0.046	0.077	0.602	0.548
H3.3	P. Dependence Barrier	−0.234	−0.233	0.075	3.139	0.002 **
H3.4	Tech. Anxiety Barrier	−0.167	−0.164	0.079	2.116	0.035 *
H3.5	Inertia (SQB)	−0.110	−0.072	0.127	0.869	0.385
**Data Management**						
H4.1	Value	−0.444	−0.343	0.290	1.534	0.125
H4.2	Usability Barrier (Rev)	−0.233	−0.235	0.049	4.724	0.000 ***
H4.3	Security Risk Barrier	−0.092	−0.091	0.057	1.604	0.109
H4.4	Tech. Anxiety Barrier	−0.172	−0.171	0.061	2.798	0.005 **

*Notes*: (Rev) = reverted scale of variable; P. = personal; Tech. = technology; SQB = status-quo barrier; M = mean; SD = standard deviation; *t*/*p* = *t*-/*p*-value, * *p* < 0.05, ** *p* < 0.01, *** *p* < 0.00.

## Data Availability

The data set can be downloaded at Appendix A.

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
