# Peer review of "Adoption of Digital Vaccination Services: It Is the Click Flow, Not the Value—An Empirical Analysis of the Vaccination Management of the COVID-19 Pandemic in Germany"

_vaccines, 2023, doi:10.3390/vaccines11040750_

Round 1

Reviewer 1 Report

Interesting work, A well constructed and a well written paper. A relevant, clear and good structured article.  Given that all readers are not so familiar with the computed values and with the statistical tests the explanation of the abbrevations and the used values should be added on the tables and the different figures in the legend of the tables and figures. it might be easier to read and to understand for the readers. Figure 1 is very difficult to read due to the colour or usde the font size.  Figure 2 which shows and interacts with the conclusion needs also some extra explanation in the legend. what is the significant factor? it might be interesting to limit the extent of your inroduction. now it is quite ample and detailed. maybe over detailed. 

Author Response

Dear Reviewer 1,

thank you very much for your review. We detected 4 improvement points from your review. Three we could directly chance, one we are not sure if it needs to be edited. For transparency reasons, we also listed the improvement points from the other reviewer. Attached you find the document.

Kind regards

The authors 

Reviewer 2 Report

Comment for MS vaccines-2291699

I would like to express my gratitude for the opportunity to review the manuscript “Adoption of digital vaccination services: It is the click flow, not the value. An empirical analysis of the vaccination management of the COVID-19 pandemic in Germany.”

Comment on this research, the researchers used the term "click flow" to refer to the process by which a user uses a mouse or an Internet button to access a web page or application that provides digital vaccination services. The purpose of this analysis is to find the most effective ways to promote the use of digital vaccination services by developing a process that is convenient and suitable for service users.

The research has clearly set up a model. and identifies the hindrance factors that may be obstacles to vaccine management, both before, such as booking appointments, appointment scheduling, and receiving injection services, as well as follow-up and subsequent management and service.

 Management and service delivery depends on the model and process in the platform, which is a technical matter (In the case of Germany). Research results suggest that usability is important because it makes the platform easy to use. There should be some brief information included What are the capabilities of this platform? And should present the deep detail of the result.

 This research aims to analyze the adoption of digital services for COVID-19 vaccination in Germany by using a survey of the states with the highest vaccination rates. To understand existing and future factors and increase vaccination efficiency, this study analyzes adaptations and barriers to adopting this digital service.

An interesting thing to do is that the author should compare the result between this state and the state where the percentage of vaccination is low or populations that are hesitant to vaccinate. To identify problems and obstacles to improving vaccination rates.

 The model setting, hypothesis formulation, and clear linkage establishment are well-defined, which reflects the questionnaire approach and can find the Significant. As shown in the picture, it is easy to understand by using just four hundred sample questionnaires.  

The results of the analysis have clear conclusions regarding:

1.      Usability affects user satisfaction.

2.      Personalization software should be tailored to make users more likely to use digital vaccination services. Access to vaccination information and the ability to book appointments quickly and conveniently quickly and conveniently are also crucial factors.

3.      Inertia, a significant factor of resistance that mainly stems from social trends.

Major concerns.

1. Please add this study's EC/IRB approval number to the manuscript to clear the state of ethical consideration.

2. This study focuses on the e-health service, including personal information with health condition states. Did this study comply with the GDPR?

3. Table 1. Why does this study not collect a certain age of participants during a survey?

Does this study include participants who were under 18 years old?

4. This study emphasises the vaccine at the state or national levels, but the sample size seems too low (eligible for only 404 responses).

Are sample sizes enough for the construction of the model?

Minor concerns.

1. Lines 50-51, "(4) vaccine effectiveness (e.g., immune response, safety, surveillance), (5) side effects (e.g., immunisation, antibody response)".

The details in parentheses may be wrong. Safety and surveillance should be in group 5, and immunisation and antibody response should be in group 4.

Comments.

1. Although the digital tool is effective in managing data. But most of the modern platform's population may be suitable to people with digital literacy, such as the new generation.

The elderly and people living in rural areas; or developing countries may struggle to participate in this platform. These groups are vulnerable to accessing a digital health service and have become neglected for e-health services.

2. In Line 70, you may use "eHealth" or "e-health" to emphasise and make it distinguishable of this term.

Misspell/typos

1. Line 46, "5'070". Suggests using a comma as a thousand separator to make it consistent with the English format.

In conclusion, this research aimed to study the creation of a model for the adoption and resistance of the digital service technology in vaccination management or digital health services in general by using web-based application technology. The areas of customization that affect usability are personalization, communication, and data management. However, there are also factors related to user ability and psychological factors that affect decision-making in technology use, particularly usability obstacles. Some factors may not be very important since technology use may occur in abnormal situations.

The research will provide guidance for managers and policymakers in crisis situations of the pandemic to have a clear path to establishing communications and improving the usability of digital services. They should prioritize click-through and interaction between servers and users, which may differ from the original factors found in traditional consumer markets that provide value for product pricing.

Finally, this research may help in developing software that meets user’s criteria, which is user-friendly, confidently accessible, safe, reliable, and matches individual attitudes, which is a principle of software engineering, then applied with vaccine management.

Author Response

Dear Reviewer 2,

thank you very much for your review. We detected 11 improvement points from your review which we could directly change or answer. For transparency reasons, we also listed the improvement points from the other reviewer. Attached you find the document.

Kind regards

The authors 
